# The mitochondrial genome of the semi-slug *Omalonyx unguis* (Gastropoda: Succineidae) and the phylogenetic relationships within Stylommatophora

**Leila Belén Guzmán**👤*, **Roberto Eugenio Vogler**👤, **Ariel Aníbal Beltramino**👤*

Grupo de Investigación en Genética de Moluscos (GIGeMol), Instituto de Biología Subtropical (IBS), CONICET–UNaM, Posadas, Misiones, Argentina

* leilaguzman95@gmail.com (LBG); beltraminoariel@hotmail.com (AAB)

**Data Availability Statement:** All relevant data are within the manuscript and its Supporting Information files.

## Abstract

Here we report the first complete mitochondrial genome of the semi-slug *Omalonyx unguis* (d'Orbigny, 1836) (Gastropoda: Succineidae). Sequencing was performed on a specimen from Argentina. Assembly was performed using Sanger data and Illumina next generation sequencing (NGS). The mitogenome was 13,984 bp in length and encoded the 37 typical Metazoan genes. A potential origin for mitochondrial DNA replication was found in a non-coding intergenic spacer (49 bp) located between *cox3* and *tRNA-Ile* genes, and its secondary structure was characterized. Secondary structure models of the tRNA genes of *O. unguis* largely agreed with those proposed for other mollusks. Secondary structure models for the two rRNA genes were also obtained. To our knowledge, the *12S-rRNA* model derived here is the first complete one available for mollusks. Phylogenetic analyses based on the mitogenomes of *O. unguis* and 37 other species of Stylommatophora were performed using amino acid sequences from the 13 protein-coding genes. Our results located Succineoidea as a sister group of Helicoidea + Urocoptoidea, similar to previous studies based on mitochondrial genomes. The gene arrangement of *O. unguis* was identical to that reported for another species of Succineoidea. The unique rearrangements observed for this group within Stylommatophora, may constitute synapomorphies for the superfamily.

## Introduction

Metazoa mitochondrial (mt) genome is a circular double stranded DNA that typically encodes two ribosomal RNAs (rRNAs), 22 transfer RNAs (tRNAs) and 13 protein-coding genes (i.e., *cytochrome c oxidase subunits I-III*; *cytochrome b*; *ATP synthase subunits 6* and *8*; *NADH dehydrogenase subunits 1–6* and *4L*), with a size ranging from 14 to 20 kb [1–4]. In recent years, the development of new technologies and the decrease in sequencing costs produced a significant increase in available animal mitogenomes, and currently more than 90,000 mitochondrial genomes are deposited in NCBI database [5–7]. Despite this and considering that mollusks are the second most specious animal phylum, they continue to be a poorly represented taxon with only 1,440 known mitogenomes (approximately 0.04% of species represented) [8].

**Funding:** The authors received no specific funding for this work.

**Competing interests:** The authors have declared that no competing interests exist.

Unlike vertebrates with a highly conserved mitochondrial genome, the available mollusk mitochondrial genomes have shown exceptional features, including doubly uniparental inheritance, gene rearrangements, large sizes, gene duplications, different gene distribution between DNA strands, and a high degree of mtDNA variability [1, 9–11]. The mtDNA as a molecular marker has been widely used throughout animal groups for the study of phylogenetic relationships among taxa, resolution of taxonomic controversies, and population genetics [12–15]. In this sense, complete mitochondrial genomes as phylogenetic markers provide additional comparison features, including gene order and content, as well as structural and compositional features [1, 16]. In addition, several studies using gene order as a phylogenetic marker indicated a good level of resolution in phylogenetic relationships, because of its low probability of convergence [3, 17, 18].

Succineidae Beck, 1837 is a molluscan family included in the superfamily Succineoidea Beck, 1837 within the order Stylommatophora that comprise more than 30 genera. The members of the family are distributed worldwide and inhabit diverse environments [19, 20]. *Omalonyx* d'Orbigny, 1838 is a genus of succinids endemic to Central and South America and the Caribbean islands characterized by a significant reduction of the shell [19]. This group of semi-slugs is frequently found in backwater areas, on riparian vegetation, with some species considered to be a pest of some crops (e.g., *Nymphoides indica*, *Pennisetum purpureum*) [19, 21–23]. They are also natural and potential intermediate hosts for parasites of the genus *Leucochloridium* Carus, 1835 (Trematoda) and *Angiostrongylus* Kamensky, 1905 (Nematoda), respectively [24–27]. Based on morphological characteristics of the reproductive system, six species are currently recognized within *Omalonyx* [28, 29], including *Omalonyx unguis* (d'Orbigny, 1836) which inhabits the Paraná River basin [28, 30].

To date, phylogenetic studies in mollusks based on morphological and molecular data support Stylommatophora as a monophyletic group, and agree to distinguish two groups, "achatinoid" and "non-achatinoid" [31–33 and references therein]. Within the non-achatinoid group, the taxonomic position and phylogenetic affinities of succinids has been variable and still remains contentious [19, 31–35].

In this study we present the mitogenome of *Omalonyx unguis* from Argentina, the first mitogenome of a South American representative of Succineidae available for comparison, increasing the taxon sampling of succinids in the ongoing phylogenetic reconstructions of Stylommatophora based on complete mitogenomes. We provide structural and compositional features of the newly sequenced mitochondrial genome, as well as an update of Stylommatophora phylogenetic relationships inferred from complete mitochondrial genomes.

## Materials and methods

### Sample and DNA extraction

Five adult specimens of *Omalonyx unguis* were collected from Garupá Stream, Misiones Province, Argentina (-27.4786, -055.7933) at the Paraná River basin in 2017. Permission for collection was granted by Ministerio de Ecología y Recursos Naturales Renovables de la Provincia de Misiones (Disp. No. 027/2018). The individuals were relaxed in water with menthol crystals for 6 h, preserved in ethanol 96%, and deposited in the malacological collection at the Instituto de Biología Subtropical, CONICET–UNaM, Misiones Province, Argentina (IBS-Ma 073). The specimens were identified using morphological characters of the reproductive system [30, 36, 37]. Additionally, molecular data were used to confirm the taxonomic identity following Guzmán et al. [30]. For the mitochondrial genome amplification and sequencing, total genomic DNA was extracted from a portion of pedal muscle of a single specimen (IBS-Ma 073–3) using a cetyltrimethyl-ammonium bromide (CTAB) protocol [38].

 2 / 22

## Amplification and sequencing

Two strategies were combined to obtain the mitochondrial genome of *Omalonyx unguis*. Firstly, short PCR products (less than ~1.5 kb) were amplified and sequenced by Sanger sequencing, using pulmonate-specific primers pairs [39] and individual-specific primers pairs designed from previously sequenced regions (S1 Table). PCR reactions were performed in a total volume of 30 μl, containing 50–100 ng of template DNA, 0.4 M of each primer, 1× *Dream Taq* Green Buffer (ThermoScientific), 2 mM $MgCl_2$, 0.2 mM dNTPs, 0.32 mg/ml BSA, and 1.25 U *Dream Taq* DNA polymerase (ThermoScientific). PCR reactions were run on a T21 thermocycler (Ivema Desarrollos) with cycling conditions as follows: 5 cycles of initial denaturation at 94°C for 2 min, denaturing at 94°C for 40 s, annealing at 40°C for 45 s, and extension at 72°C for 1 min; followed by 30 cycles of denaturing at 94°C for 40 s, annealing at 40–60°C for 40 s (based on gradient PCR profile), and extension at 72°C for a time set at 1 minute per kb of expected product, with a final extension at 72°C for 3 min. PCR products were purified from solution by means of an AccuPrep PCR Purification Kit (Bioneer, Korea), and from 1.5% (w/v) agarose gels using an ADN PuriPrep-GP Kit (Inbio Highway, Argentina) in cases of co-amplification of nonspecific fragments. Finally, both DNA strands of the PCR products were directly cycle sequenced (Macrogen Inc., Seoul, Korea). Secondly, NGS was used for sequencing of the mitogenomic fragments. Paired-end sequencing (2x150 bp, 350 bp insert size) of total genomic DNA was performed by Novogene Corporation (Sacramento, CA, USA) using a HiSeq platform (Illumina).

## Genome assembly and annotation

All fragments obtained by Sanger sequencing were edited and compared with reference sequences in GenBank using the BLASTn algorithm [40] to confirm the amplicons to be the target sequences. Then, sequences were assembled manually into two large contigs in a stepwise manner by concatenating the sequences and trimming off overlapping regions with the help of Bioedit 7.0.5 software [41]. Finally, the complete mitogenome was assembled from about 10 Gb raw data obtained by NGS with the software NOVOPlasty 3.8.2 (https://github.com/ndierckx/NOVOPlasty) [42] using the previously obtained contigs as starting seeds.

The mitogenome was annotated with MITOS Web Server [5] employing the invertebrate mitochondrial genetic code and corrected manually by comparison with the available mitochondrial genome for the confamilial species *Succinea putris* (Linnaeus, 1758) (JN627206). The tRNA genes were detected using MITOS, and manually checked. Additionally, their secondary structures were inferred with ARWEN 1.2 [43]. The protein-coding genes (PCGs) were validated using the NCBI ORF Finder resource (https://www.ncbi.nlm.nih.gov/orffinder/). Finally, the limits of rRNAs were extend to the boundaries of adjacent genes following Cameron [6]. Secondary structure of the *16S-rRNA* and *12S-rRNA* genes was predicted from previous models for mollusks [44] and arthropods [14, 45], respectively, with the help of RNAStructure 6.0.2 (https://rna.urmc.rochester.edu/RNAstructureWeb) [46]. Additionally, the potential origin of DNA replication (POR) folding was performed using RNAStructure. The annotated mitogenome was deposited in GenBank under accession number MT449229. The nucleotide composition and relative synonymous codon usage (RSCU) for PCGs were calculated by means of MEGA X [47]. The AT- and GC- skew values were calculated using the equations AT-skew = (A-T)/(A+T), and GC-skew = (G-C)/(G+C) [48]. The circular mitochondrial genome was generated using Mtviz online tool (http://pacosy.informatik.uni-leipzig.de/mtviz/).

## Sequence alignments and phylogenetic analyses

Phylogenetic analyses were based on the 13 PCGs of *O. unguis* and 37 species of Stylommatophora available in GenBank (Table 1). *Galba pervia* (Martens, 1867) (Hygrophila), *Platevindex mortoni* Britton, 1984 (Systellommatophora) and *Carychium tridentatum* (Risso, 1826) (Ellobioidea) were used as outgroups (Table 1). All sequences were downloaded from NCBI with the R package AnnotationBustR [49]. Nucleotide sequences were translated into amino acid sequences in EMBOSS Transeq [50] with invertebrate mitochondrial code; subsequently, sequences of each protein-coding gene were aligned separately using Muscle as implemented in MEGA X [47]. Ambiguously aligned positions were removed using Gblocks 0.91b under relaxed settings [51]. Exceptionally, *atp8* and *nad4L* sequences were manually cleaned. Finally, the single alignments were concatenated into a final dataset consisting of 2,877 positions (71% of the original alignment).

Phylogenetic trees were inferred using Bayesian Inference (BI) and Maximum Likelihood (ML) following Uribe et al. [52] with some modifications. BI analysis was performed using MrBayes on XSEDE 3.2.6 [53] implemented in the CIPRES Science Gateway [54]. The substitution model was determined by PartitionFinder 1.1.1 [55] according to the Bayesian Information Criterion (BIC) using a greedy approach. The phylogenetic analysis was performed under the site-homogeneous mtREV+I+G model. Two independent runs with four Markov chains were set to run simultaneously for $10^6$ generations, sampling every 1,000 generations, with a final burn-in of 25% [56]. Bayesian posterior probabilities (PP) were used as branch support values. ML analysis was conducted using IQ-TREE 1.6.12 [57]. The substitution models were estimated using ModelFinder [58] implemented in IQ-TREE. ML analysis was performed under the site-homogeneous mtZOA+F+R7 model. Additionally, nodal support values were evaluated with 10,000 replicates of ultrafast likelihood bootstrap (UFBoot) [59]. Finally, as particular Heterobranchia lineages are known to experience an acceleration of evolutionary rates that may lead to long-branch attraction artifacts (LBA) in phylogenies, an additional BI analysis using the site-heterogeneous CAT-GTR model was performed in PhyloBayes MPI 1.7b [60] implemented in the CIPRES Science Gateway following Uribe et al. [61]. The phylogenetic trees were visualized using FigTree 1.4.4 (http://tree.bio.ed.ac.uk/software/figtree/).

## Results

### Genome organization and features

The complete mitochondrial genome of *Omalonyx unguis* was 13,984 bp in length, and contained 13 PCGs, 22 tRNAs, and two RNA genes (Fig 1). Most genes (9 PCGs, 12 tRNAs, 1 rRNA) were found on the plus strand, while the remaining genes (4 PCGs, 10 tRNAs, 1 rRNA) were on the minus strand (Fig 1, Table 2). We identified 17 intergenic regions (113 bp in total) ranging from 1 to 49 bp. The largest one was located between *cox3* and *tRNA-Ile* genes with an AT content of 98% (Table 2). This sequence was identified as POR and its secondary structure is shown in S1 Fig. Additionally, there were 13 overlapping regions with 4–60 bp size, and six genes pairs were directly adjacent to one another. The base composition of this genome (33.38% A, 44.07% T, 10.59% C, 11.96% G) showed a high AT content (77.5%), with a negative AT-skew value (-0.14) and positive GC-skew value (0.06). The protein-coding genes accounted for 77.1% of the mitochondrial genome of *O. unguis*, with *nad5* and *atp8* being the longest and shortest genes, respectively (Table 2). The PCGs were initiated with ATG and the alternative initiation codons TTG, ATT or ATA, and terminated with stop codons TAA, TAG or incomplete stop codon T (Table 2). In total, PCGs consisted of 3,607 codons (not including start and stop codons), among which UUU (Phe, counted 393 times), UUA (Leu, counted 397 times)

**Table 1. Information of the mitochondrial genomes of Stylommatophora analyzed in present paper.**

| Species | Family | NCBI | Mitogenome size (bp) | References |
|---|---|---|---|---|
| Stylommatophora | | | | |
| *Omalonyx unguis* | Succineidae | MT449229 | 13,984 | This study |
| *Succinea putris* | Succineidae | NC016190 | 14,092 | White et al. [39] |
| *Achatinella fulgens* | Achatinellidae | MG925058 | 15,346 | Price et al. [62] |
| *Achatinella mustelina* | Achatinellidae | NC030190 | 16,323 | Price et al. [91] |
| *Achatinella sowerbyana* | Achatinellidae | KX356680 | 15,374 | Price et al. [92] |
| *Partulina redfieldi* | Achatinellidae | MG925057 | 16,879 | Price et al. [62] |
| *Achatina fulica* | Achatinidae | KM114610 | 15,057 | He et al. [93] |
| *Deroceras reticulatum* | Agriolimacidae | NC035495 | 14,048 | Ahn et al. [94] |
| *Arion vulgaris* | Arionidae | MN607980 | 14,548 | Doğan et al. [7] |
| *Arion rufus* | Arionidae | KT626607 | 14,321 | Romero et al. [95] |
| *Aegista aubryana* | Bradybaenidae | NC029419 | 14,238 | Yang et al. [96] |
| *Aegista diversifamilia* | Bradybaenidae | NC027584 | 14,039 | Huang et al. [97] |
| *Dolicheulota formosensis* | Bradybaenidae | NC027493 | 14,237 | Huang et al. [97] |
| *Mastigeulota kiangsinensis* | Bradybaenidae | NC024935 | 14,029 | Deng et al. [68] |
| *Camaena cicatricosa* | Camaenidae | NC025511 | 13,843 | Wang et al. [98] |
| *Camaena poyuensis* | Camaenidae | KT001074 | 13,798 | Unpublished |
| *Cerion incanum* | Cerionidae | NC025645 | 15,177 | González et al. [69] |
| *Cerion tridentatum* | Cerionidae | KY249249 | 15,409 | Unpublished |
| *Cerion uva* | Cerionidae | KY124261 | 15,043 | Harasewych et al. [99] |
| *Ryssota otaheitana* | Chronidae | NC044784 | 13,888 | Damatac and Fontanilla [88] |
| *Albinaria caerulea* | Clausiliidae | NC001761 | 14,130 | Hatzoglou et al. [100] |
| *Gastrocopta cristata* | Gastrocoptidae | NC026043 | 14,060 | Unpublished |
| *Cernuella virgata* | Geomitridae | NC030723 | 14,147 | Lin et al. [70] |
| *Helicella itala* | Geomitridae | KT696546 | 13,967 | Romero et al. [95] |
| *Cepaea nemoralis* | Helicidae | NC001816 | 14,100 | Yamazaki et al. [77] |
| *Cylindrus obtusus* | Helicidae | NC017872 | 14,610 | Groenenberg et al. [101] |
| *Cornu aspersum* | Helicidae | NC021747 | 14,050 | Gaitán-Espitia et al. [4] |
| *Helix pomatia* | Helicidae | NC041247 | 14,072 | Korábek et al. [102] |
| *Orcula dolium* | Orculidae | NC034782 | 14,063 | Groenenberg et al. [103] |
| *Oreohelix idahoensis* | Oreohelicidae | NC043790 | 14,213 | Linscott and Parent [89] |
| *Naesiotus nux* | Orthalicidae | NC028553 | 15,197 | Hunter et al. [104] |
| *Meghimatium bilineatum* | Philomycidae | NC035429 | 13,972 | Xie et al. [86] |
| *Philomycus bilineatus* | Philomycidae | MG722906 | 14,347 | Yang et al. [105] |
| *Polygyra cereolus* | Polygyridae | NC032036 | 14,008 | Unpublished |
| *Praticolella mexicana* | Polygyridae | KX240084 | 14,153 | Minton et al. [106] |
| *Pupilla muscorum* | Pupillidae | NC026044 | 14,149 | Unpublished |
| *Microceramus pontificus* | Urocoptidae | NC036381 | 14,275 | Unpublished |
| *Vertigo pusilla* | Vertiginidae | NC026045 | 14,078 | Unpublished |
| Ellobiida* | | | | |
| *Carychium tridentatum* | Ellobiidae | KT696545 | 13,908 | Romero et al. [95] |
| Hygrophila* | | | | |
| *Galba pervia* | Lymnaeidae | NC018536 | 13,768 | Liu et al. [13] |
| Systellommatophora* | | | | |
| *Platevindex mortoni* | Onchidiidae | GU475132 | 13,991 | Sun et al. [107] |

* denotes outgroups.

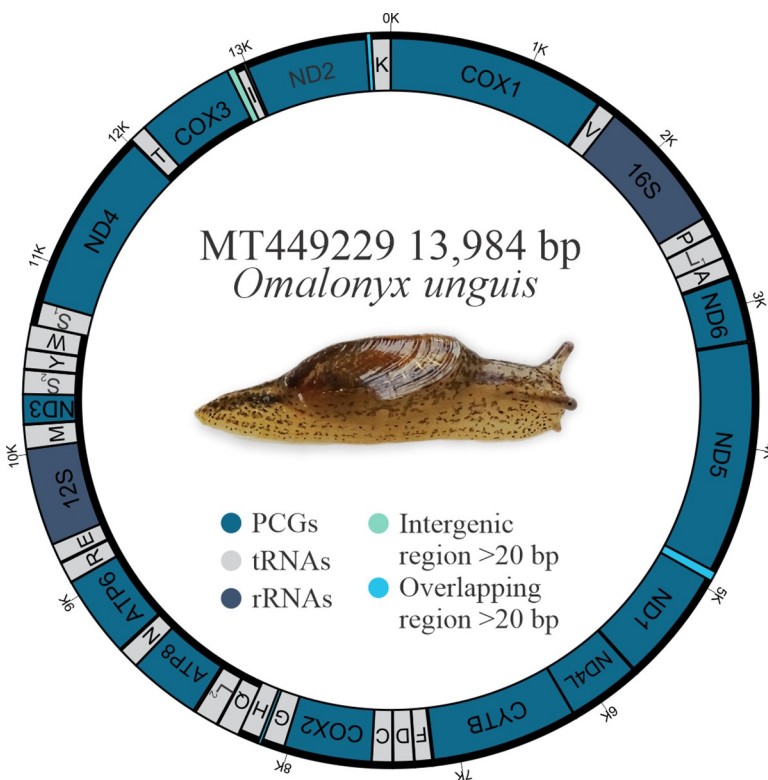

**Fig 1. Organization of the mitogenome of *Omalonyx unguis*.** Black lines outside and inside the circle indicate the plus and minus strands, respectively. The types of genes are represented by different colors. Transfer RNAs are designated by the single-letter code of their cognate amino acid.

and AUU (Ile, counted 360 times) were the most frequently used, while CCG (Pro, counted 1 time), GCG (Ala, counted 2 times) ACG, CGG, and AGC (Thr, Arg and Ser₁, respectively, counted 3 times) were the least used (S2 Table). The RSCU values are shown in Fig 2 and S2 Table. All 22 tRNAs were identified by both ARWEN and MITOS, and their length ranged from 57 to 75 bp. The tRNA genes in *O. unguis* showed a classical clover-leaf secondary structure with exception of *tRNA-Leu₂*, *tRNA-Lys*, and *tRNA-Ser₂*. The tRNA structures are shown in Fig 3. Twenty of the 22 tRNAs presented the standard anticodons, while *tRNA-Trp* and *tRNA-Lys* were identified with the anticodons UCA and UUU, respectively. In addition, seven tRNAs (*tRNA-Ala*, *tRNA-Asp*, *tRNA-His*, *tRNA-Leu₂*, *tRNA-Lys*, *tRNA-Pro* and *tRNA-Tyr*) showed mismatched base pairs in the acceptor stem and three (*tRNA-Val*, *tRNA-Trp* and *tRNA-Gly*) in the anticodon stem, while three tRNAs (*tRNA-Asp*, *tRNA-Gly* and *tRNA-Pro*) showed G·U wobble base pairs in the acceptor stem, and another two (*tRNA-Ile* and *tRNA--Ser₂*) in the anticodon stem. Finally, the lengths of the *12S-rRNA* and *16S-rRNA* genes were 771 and 1,013 bp, respectively, with the first located on the minus strand and the second on the plus strand. The inferred secondary structure models for *12S-rRNA* and *16S-rRNA* are provided in Figs 4 and 5, respectively.

## Phylogenetic analyses, gene order and rearrangements

The results of the phylogenetic reconstructions under the ML and BI approaches from concatenated amino acid sequences of the 13 PCGs and carried out using site-homogeneous models are shown in Fig 6. The topology of both trees was not entirely identical and nodal

**Table 2. Organization of the mitochondrial genome of *Omalonyx unguis*.**

| Name | Position | Length (bp) | Strand | Start Codon | Stop Codon | Anticodon | ISR (bp) |
|---|---|---|---|---|---|---|---|
| *cox1* | 1–1,530 | 1,530 | + | TTG | TAA | | +8 |
| tRNA$^{Val}$ | 1,539–1,604 | 66 | + | | | TAC | 0 |
| *16S-rRNA* | 1,605–2,617 | 1,013 | + | | | | 0 |
| tRNA$^{Pro}$ | 2,618–2,680 | 63 | + | | | TGG | -4 |
| tRNA$^{Leu1}$ | 2,677–2,740 | 64 | + | | | TAG | +2 |
| tRNA$^{Ala}$ | 2,743–2,805 | 63 | + | | | TGC | -14 |
| *nad6* | 2,792–3,268 | 477 | + | ATT | TAA | | -11 |
| *nad5* | 3,258–4,940 | 1,683 | + | ATG | TAA | | -60 |
| *nad1* | 4,880–5,827 | 948 | + | TTG | TAA | | +9 |
| *nad4L* | 5,837–6,110 | 274 | + | ATA | T* | | -3 |
| *cob* | 6,108–7,202 | 1,095 | + | ATT | TAG | | -10 |
| tRNA$^{Phe}$ | 7,193–7,257 | 65 | + | | | GAA | +4 |
| tRNA$^{Asp}$ | 7,262–7,324 | 63 | + | | | GTC | -5 |
| tRNA$^{Cys}$ | 7,320–7,385 | 66 | + | | | GCA | +1 |
| *cox2* | 7,387–8,035 | 649 | + | ATG | T* | | 0 |
| tRNA$^{Gly}$ | 8,036–8,110 | 75 | + | | | TCC | -21 |
| tRNA$^{His}$ | 8,090–8,155 | 66 | + | | | GTG | +1 |
| tRNA$^{Gln}$ | 8,157–8,221 | 65 | - | | | TTG | +7 |
| tRNA$^{Leu2}$ | 8,229–8,287 | 59 | - | | | TAA | -15 |
| *atp8* | 8,273–8,402 | 130 | - | TTG | T* | | +1 |
| tRNA$^{Asn}$ | 8,404–8,466 | 63 | - | | | GTT | -10 |
| *atp6* | 8,457–9,122 | 666 | - | ATG | TAG | | +2 |
| tRNA$^{Arg}$ | 9,125–9,189 | 65 | - | | | TCG | +8 |
| tRNA$^{Glu}$ | 9,198–9,270 | 73 | - | | | TTC | 0 |
| *12S-rRNA* | 9,271–10,041 | 771 | - | | | | 0 |
| tRNA$^{Met}$ | 10,042–10,107 | 66 | - | | | CAT | +4 |
| *nad3* | 10,112–10,465 | 354 | - | TTG | TAA | | +1 |
| tRNA$^{Ser2}$ | 10,467–10,523 | 57 | - | | | TGA | -6 |
| tRNA$^{Tyr}$ | 10,518–10,584 | 67 | - | | | GTA | +5 |
| tRNA$^{Trp}$ | 10,590–10,655 | 66 | - | | | TCA | 0 |
| tRNA$^{Ser1}$ | 10,654–10,716 | 63 | + | | | GCT | +1 |
| *nad4* | 10,718–12,028 | 1,311 | + | TTG | TAA | | -8 |
| tRNA$^{Thr}$ | 12,021–12,084 | 64 | - | | | TGT | +1 |
| *cox3* | 12,086–12,865 | 780 | - | ATG | TAG | | +49 |
| tRNA$^{Ile}$ | 12,915–12,984 | 70 | + | | | GAT | +16 |
| *nad2* | 13,001–13,958 | 958 | + | TTG | T* | | -33 |
| tRNA$^{Lys}$ | 13,926–13,983 | 58 | + | | | TTT | +1 |

+ and—denote plus and minus strands, respectively. ISR denotes the length of the intergenic spacer region, for which negative numbers indicate nucleotide overlapping between adjacent genes. The anticodons of tRNAs are reported in the 5' - 3' direction.

* denotes incomplete stop codon.

support values were generally higher in BI than in ML. Both approaches recovered all super-families as monophyletic groups with high support values and distinguished the "non-achatinoid" group from the "achatinoid" one represented by *Achatina fulica* Bowdich, 1822 (the only mitogenome available for achatinoids). In this study, the Succineoidea (represented by the succinids *Omalonyx unguis* and *Succinea putris*) showed a sister group relationship to

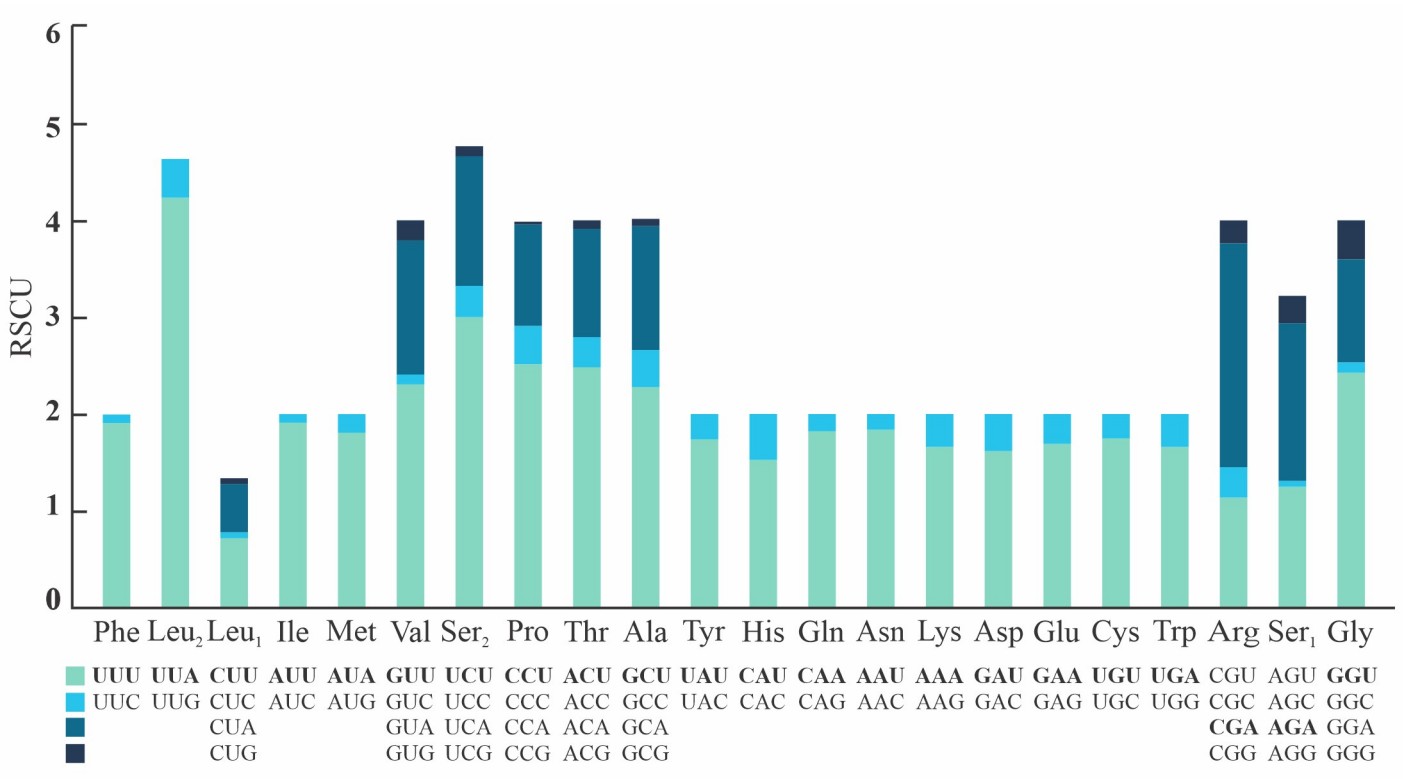

**Fig 2. Relative synonymous codon usage (RSCU) of the mitogenome of *Omalonyx unguis*.** The codons that compose each family are shown below the x-axis, and the colors correspond to those of the stacked columns. The RSCU values are shown on the y-axis. The biased codon for each amino acid family is highlighted in bold.

Helicoidea + Urocoptoidea with relatively high nodal support (UFBootS 78, PP 0.98). Although *Naesiotus nux* (Broderip, 1832) and *Oreohelix idahoensis* (Newcomb, 1866) grouped together with high support values (Orthalicoidea + Punctoidea), the position of this group differed between the two approaches. On the other hand, there were differences in the relationships established among families within Helicoidea when comparing both trees. Helicidae grouped together with Geomitridae in both approaches, however, Polygyridae grouped together with Camaenidae in the BI tree (Camaenidae + Polygyridae) + (Helicidae + Geomitridae), but they grouped outside of Camaenidae + (Helicidae + Geomitridae) in the ML tree. The phylogenetic tree of the BI analysis performed under the site-heterogeneous model is shown in S2 Fig. The tree topology obtained using the CAT-GTR model, with branches showing high support values, was similar to that of the BI tree under the site-homogeneous MTREV +I+G model (Fig 6B).

Regarding the order and orientation of mitochondrial genes, some differences were observed among the species of the different stylommatophoran families (Fig 7). In particular, *O. unguis* presented a gene arrangement identical to that reported for *S. putris*. Both succinids differed from the other stylommatophoran by exhibiting the following rearrangements: *tRNA-Pro–tRNA-Leu₁–tRNA-Ala*, *tRNA-Phe–tRNA-Asp–tRNA-Cys*, and *tRNA-Ser₂–tRNA--Tyr–tRNA-Trp–tRNA-Ser₁* (with *tRNA-Tyr* and *tRNA-Trp* on the minus strand).

## Discussion

The mitogenome of the semi-slug *Omalonyx unguis* obtained in this study presented a size of 13,984 bp and constitutes the second complete mitochondrial genome available for

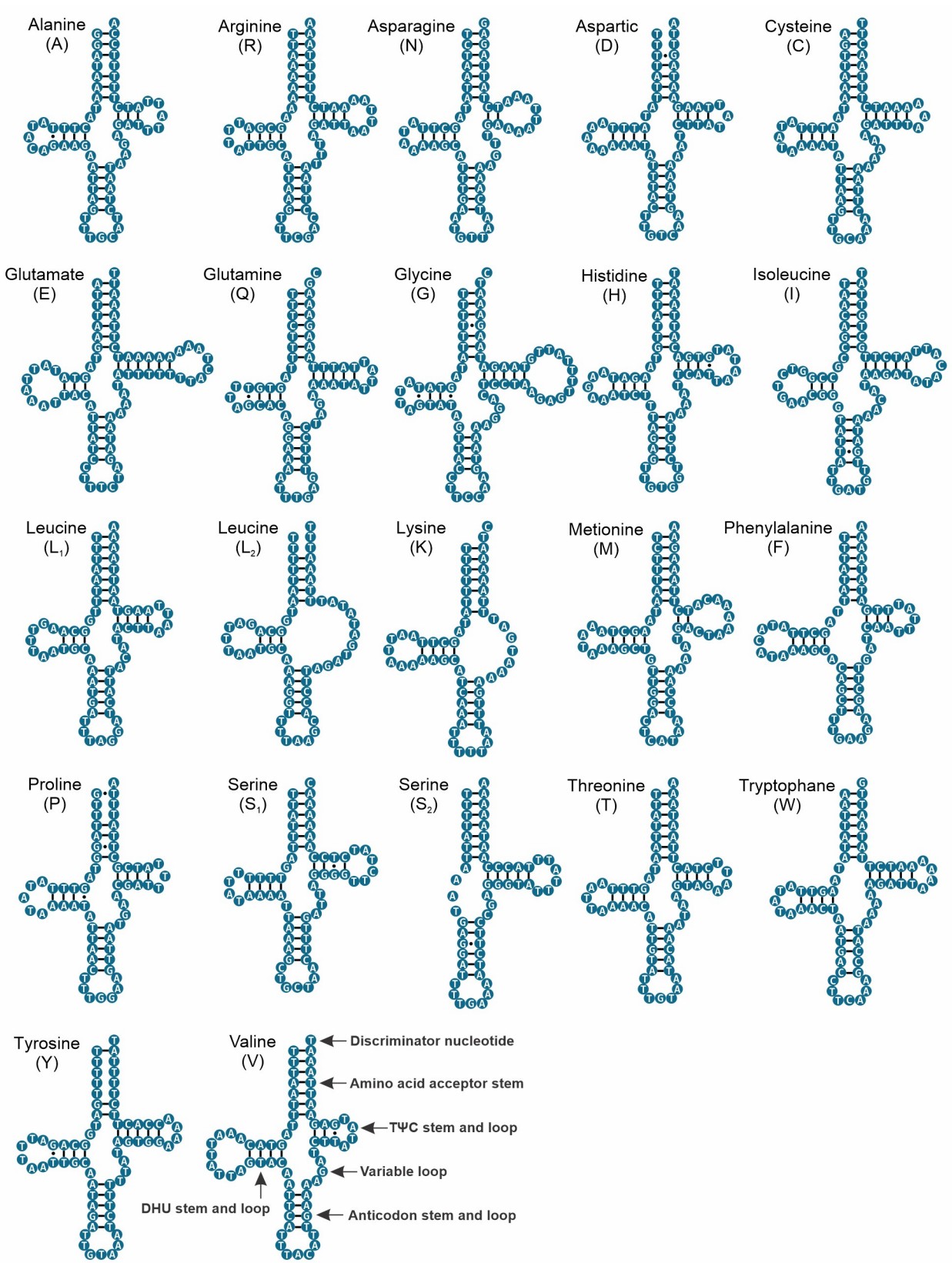

**Fig 3. Putative secondary structure of the 22 tRNA genes identified in the mitogenome of *Omalonyx unguis*.** The tRNAs are labeled with the abbreviations of their corresponding amino acids. Watson-Crick base pairings are indicated by dashes (−), and G·T(U) wobble base pairings are indicated by dots (·).

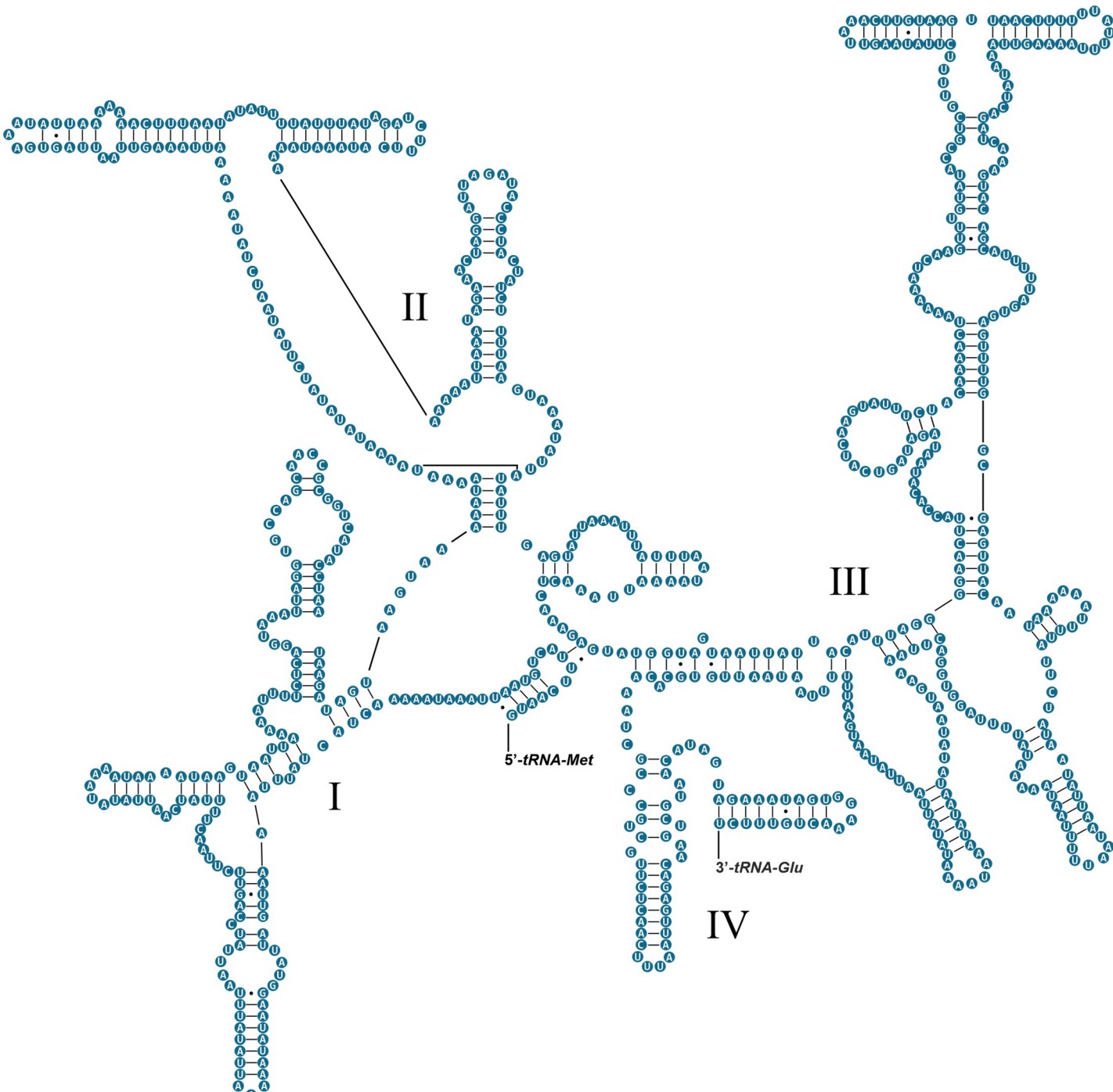

**Fig 4. Secondary structure of the *12S-rRNA* gene of *Omalonyx unguis*.** Domains are indicated with Roman numbers. Watson-Crick base pairings are indicated by dashes (−), and G·U wobble base pairings are indicated by dots (·). Adjacent genes are labelled at the 5' and 3' ends.

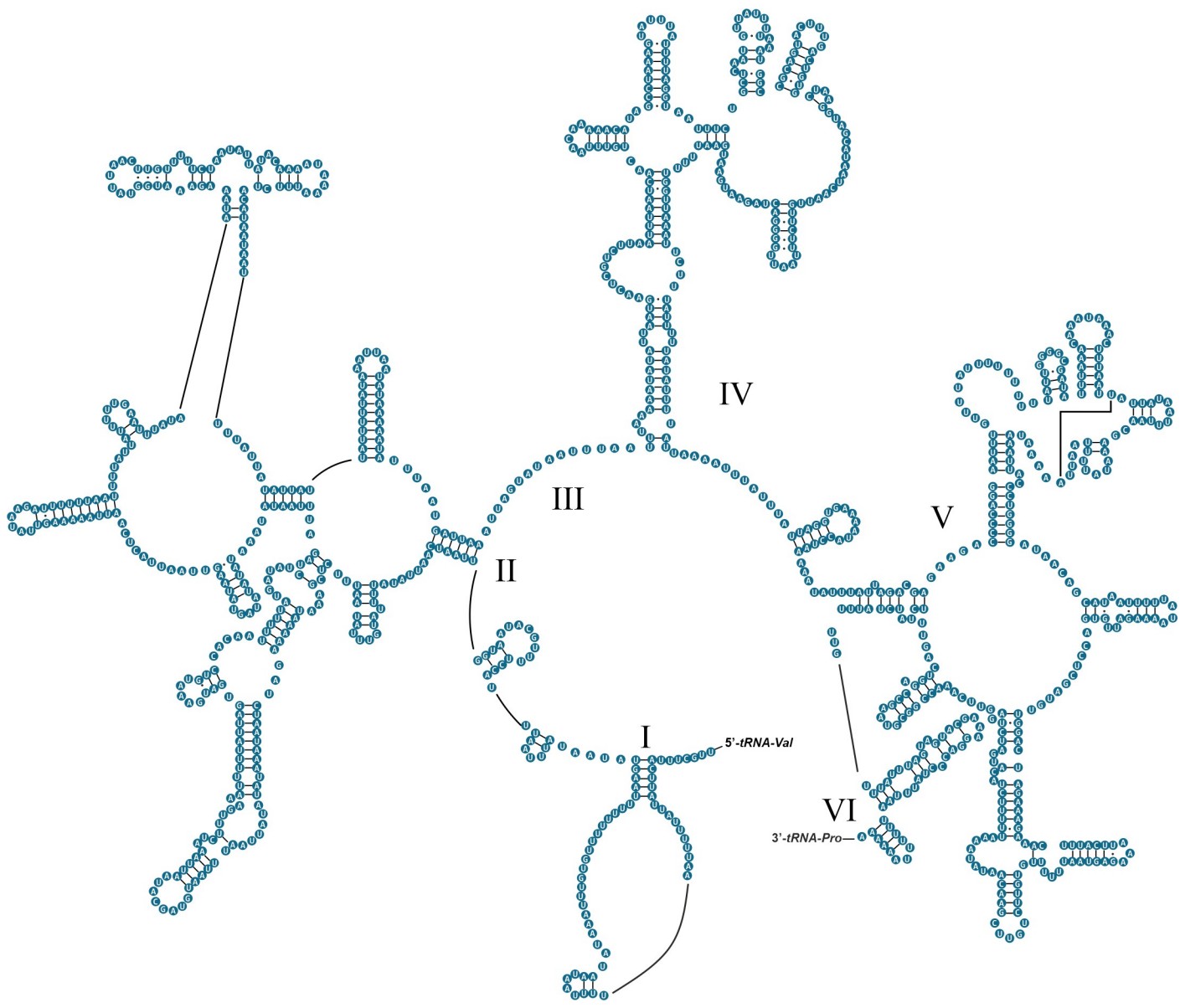

**Fig 5. Secondary structure of the *16S-rRNA* gene of *Omalonyx unguis*.** Domains are indicated with Roman numbers. Watson-Crick base pairings are indicated by dashes (−), and G·U wobble base pairings are indicated by dots (·). Adjacent genes are labelled at the 5' and 3' ends.

Succineidae worldwide. The reduced size of this genome is consistent with that reported for Stylommatophora, which account for 38 mitogenomes to date with sizes ranging from 13,978 bp for *Camaena poyuensis* Zhou et al., 2016 to 16,879 bp for *Partulina redfieldi* (Newcomb, 1853) [2, 3, 62]. The 37 typical genes (13 PCGs, 22 tRNAs and 2 rRNAs) were present, and contained 17 intergenic spacer regions and 13 overlapping regions. Origins of mitochondrial replication are reported to be characterized by high AT content and stem-loop structures [63, 64]. These features were observed in the longest intergenic region (49 bp) of the *O. unguis* mitogenome located between *cox3* and *tRNA-Ile* genes. The location of a potential origin of DNA replication (POR) in this region is in line with what has been observed in *S. putris* and other heterobranchs [3, 4, 39]. The compositional approach showed a high AT content

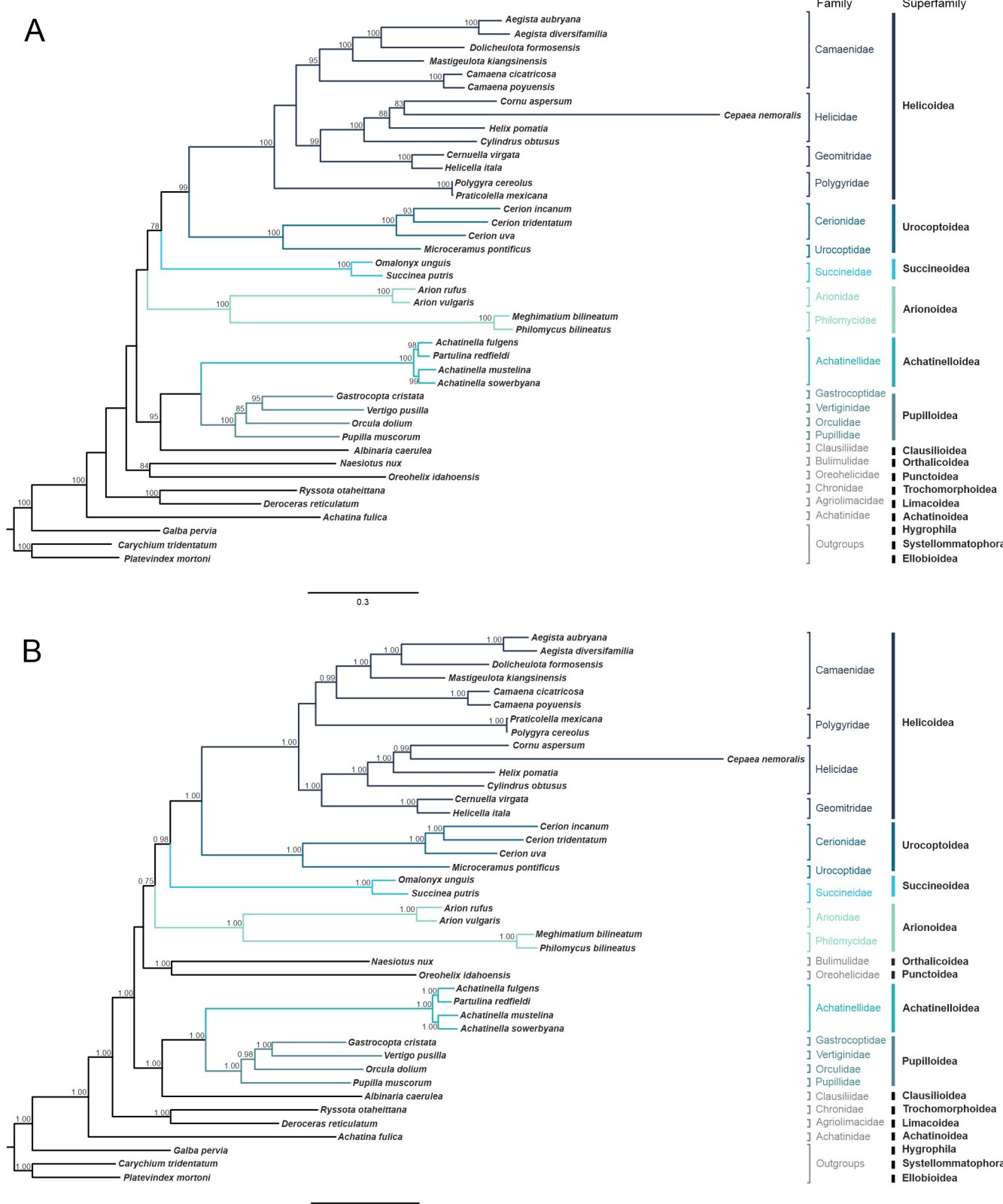

**Fig 6. Phylogenetic trees under site-homogeneous models based on the 13 PCGs dataset for stylommatophoran mollusks.** A. Maximum Likelihood tree. B. Bayesian consensus tree. The trees were rooted with three outgroups (*Carychium tridentatum*, *Platevindex mortoni* and *Galba pervia*). The scale (0.3)

shows evolutionary distances. The ultrafast bootstrap (ML) and posterior probability (BI) support values are shown in the nodes. The GenBank accession numbers of the species are shown in Table 1. *Omalonyx unguis* was sequenced in this study.

(77.5%) in *O. unguis*, with a value close to that described for the mitochondrial genome of *S. putris* (76.7%). This high content fits well with the expectations for Stylommatophora, which stand out for having the highest values of AT among the pulmonate gastropods [4]. Generally, the plus chain is characterized by a higher A and C content [65]. Within Mollusca, two groups have been characterized based on their compositional asymmetries. On one hand, cephalopods

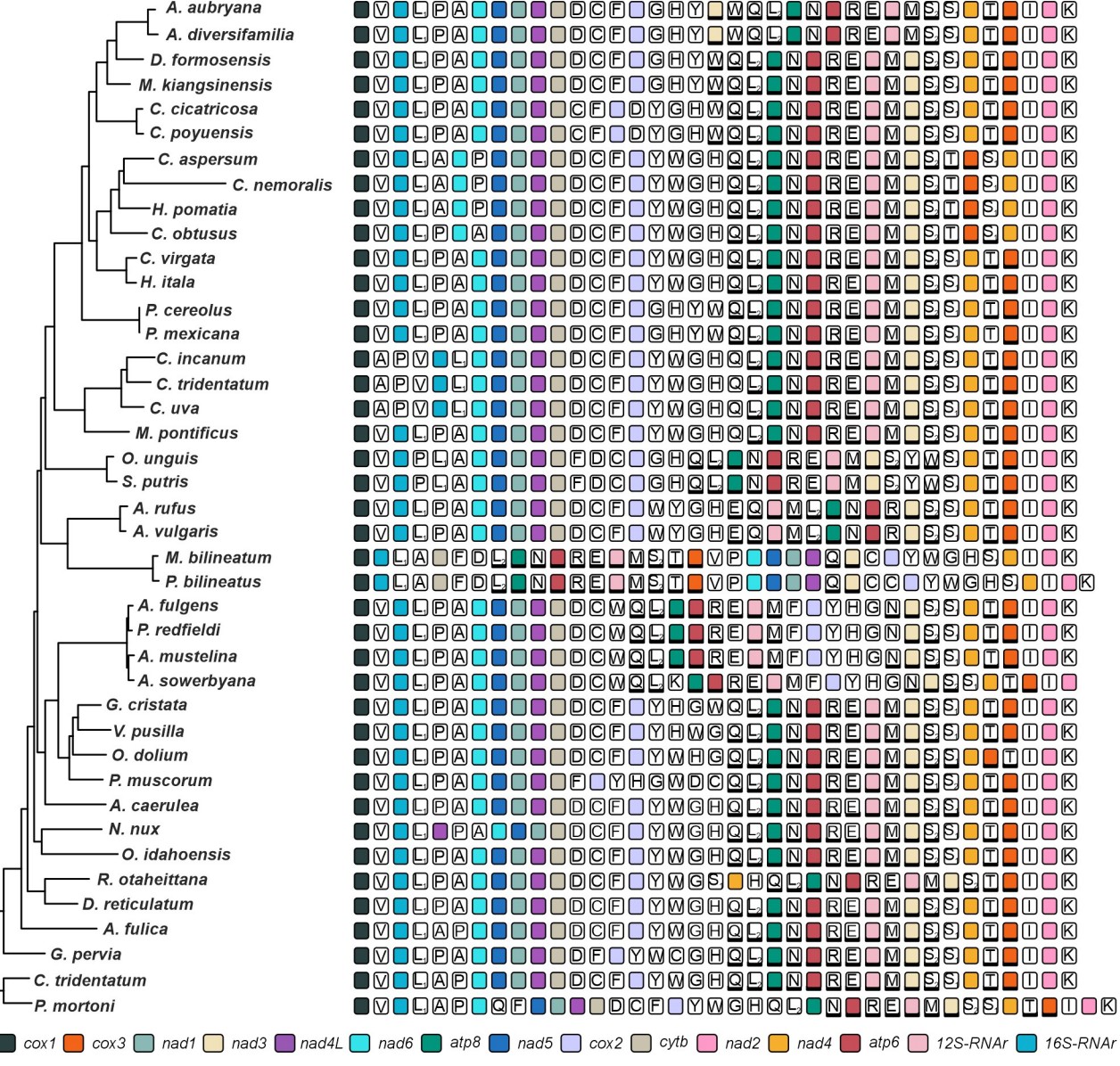

**Fig 7. Linear representation of the gene order in Stylommatophora species used in this study.** Tree topology is from Maximum Likelihood analysis. Genes encoded by the minus strand are shaded. tRNAs are designated by the single-letter code of their cognate amino acid: A, Ala; C, Cys; D, Asp; E, Glu; F, Phe; G, Gly; H, His; I, Ile; K, Lys; L₁ and L₂, Leu; M, Met; N, Asn; P, Pro; Q, Gln; R, Arg; S₁ and S₂, Ser; T, Thr; V, Val; W, Trp; Y, Tyr.

and some gastropod species with positive AT and negative GC biases, and on the other hand, bivalves, and most gastropods with inverse values [65]. In agreement with most gastropods and other animal groups such as fish and arthropods, *O. unguis* showed a strong inversion of its asymmetries (AT-skew = -0.14, GC-skew = 0.06) [64, 66]. Some authors have suggested that one of the reasons for this finding could be related to a reversal of the origin of the replication [64, 66, 67].

The size of the protein-coding genes in *O. unguis* conformed well to the expected pattern for Eupulmonata, with *nad5* being the longest gene (1,683 bp) and *atp8* the shortest one (130 bp) [4]. The start and stop codons of the *cox2*, *cox3* and *nad5* genes corresponded to those identified for *S. putris* [39]. For the remaining genes, some start or stop codons matched with those of *S. putris*, while others corresponded to those informed for other pulmonate gastropods [3, 4, 39, 68–70]. Exceptionally, the *nad4L*, *cox2*, *nad2* and *atp8* genes showed a truncated stop codon (T—). This type of codon has been widely documented for mollusks and it was suggested that it is completed to TAA by post-transcriptional polyadenylation [71, 72]. In relation with mitogenomes that exhibited a high AT content, some Metazoa revealed a strong impact on the use of codons with this composition, even leading to the extinction of codons with GC content [73]. Accordingly, the analysis of codon uses in *O. unguis* revealed a higher use of those with high AT content, such as TTT (393), TTA (397), ATT (360), while those less represented had high GC content. Despite this, all possible codons were present at least once, as was the case with the CCG codon. The two most represented codon families were $Ser_2$ and $Leu_2$ with RSCU values of 4.77 and 4.65 respectively, while the least prevalent was $Leu_1$ with a value of 1.35. The compositional bias in *O. unguis* was reflected in the RSCU values that showed no random use among families of codon synonyms, since a preferential use of codons with A or T in their third position was observed. It is known that codon bias influences the folding and differential regulation of proteins, as well as the efficiency of translation [74], however, to our knowledge the impact it has on gastropods mitogenomes and how it varies among its taxa is still unknown.

Recurrently, the variability in the sequence of tRNA genes generates difficulties for sequence annotation, so secondary structures are especially useful accessory tools [75]. tRNAs are generally characterized by a cloverleaf-shaped secondary structure that adopts an L-shaped tertiary structure to perform its function [76]. Most of the tRNAs of *O. unguis* showed the typical shape except for *tRNA-Leu₂* and *tRNA-Lys* which showed the TΨC arm truncated, and *tRNA-Ser₂* which showed the DHU arm truncated. However, loss of arms (TΨC or DHU) has been described as structural variation for Metazoa, including some species of mollusks, and is suggested to be the result from selection pressure to reduce the size of mitochondrial genomes [3, 18, 77, 78]. Most of the characterized structures showed preserved sizes for the acceptor stem and the anticodon stem (7 bp and 5 bp, respectively), as expected for metazoans in general [70, 79]. However, the acceptor stem of the *tRNA-Ala*, *tRNA-Asp*, *tRNA-His*, *tRNA-Leu₂*, *tRNA-Lys*, *tRNA-Pro* and *tRNA-Tyr*, and the anticodon stem of the *tRNA-Val*, *tRNA-Trp* and *tRNA-Gly* showed mismatched base pairs. Exceptionally, *tRNA-Ser₂* showed a longer length than conventional in the anticodon stem (6 bp). These types of alterations have been frequently observed in other animal species involving diverse mitochondrial tRNAs [72, 77, 80]. Particularly for mollusks, Yamazaki et al. [77] concluded that tRNAs that overlap nucleotides with adjacent genes may have a lack of mating in the acceptor stem. This was observed in *O. unguis* (e.g., *tRNA-Ala*, *tRNA-Tyr* and *tRNA-Leu₂*), although most tRNAs overlapping nucleotides were unmodified. To ensure the functionality of tRNAs that show modifications in their pairings, some authors proposed that post-transcriptional editing is responsible for rectifying these disappearances by Watson-Crick pairings [3, 77, 81, 82]. Despite the results provided here, more mitochondrial genomes of mollusks including secondary structures of these genes

are required. These will allow comparative studies to help establish patterns of mitochondrial evolution, as well as to understand the functional importance of these structures.

The sizes of the large (*16S-rRNA*) and small (*12S-rRNA*) rRNA subunits in *O. unguis* were similar to those reported for other mollusks [39]. The *16S-rRNA* structure model showed a relatively conserved secondary structure in relation to the available models for mollusks, and presented the six typical domains [44]. However, a stem-loop structure was not present in neither domain II or III, nor a bulge-stem-loop structure was present in domain V. These features are known to represent synapomorphies described for Heterobranchia and explain some reduction in mitochondrial size [44, 83]. Additionally, we generated the secondary structure model for the *12S-rRNA* gene which, to our knowledge, represents the first complete model available for Mollusca. It was developed based on available models for arthropods and presented the four typical domains into which this gene is structurally divided [14, 72, 84]. This model is expected to contribute to further comparative studies aimed to investigate evolutionary questions among mollusks.

The BI and ML phylogenetic trees based on amino acid sequences of 13 PCGs recovered Stylommatophora as a monophyletic group. Both analyses showed a basal dichotomy within the order Stylommatophora that separated "non-achatinoid" (suborder Helicina) and "achatinoid" (suborder Achatinina), the latter represented only by *Achatina fulica*. This finding agrees well with previous studies in the literature based on mitochondrial genomes as well as individual genes, and represent widely accepted taxonomic clades [34, 85, 86]. Within Helicina, both trees reconstructed under site-homogeneous models recovered 11 superfamilies as monophyletic groups (UFBootS = 100, PP = 1.0), with similar topologies to studies carried out so far with mitogenome-based phylogenies. One of the previous mitogenomic studies grouped Succineoidea together with Arionoidea, which was interpreted as an artifact attributed to a scarce sampling of taxa [86]. Our results located Succineoidea as a sister group of Helicoidea + Urocoptoidea with Arionoidea grouping outside. These findings are consistent with the most recent results from Doğan et al. [7]. However, earlier phylogenies based on individual genes showed close relationships between Arionoidea and Limacoidea, while they failed to clarify the relationships of Elasmognatha (Succineidae + Athoracophoridae) within "achatinoids" [32]. Previous phylogenetic studies in Stylommatophora, performed on individual genes, also showed a close relationship between Polygyridae and Camaenidae within the superfamily Helicoidea [32, 87]. However, some phylogenies based on mitochondrial genomes showed Polygyridae in the basal division of Helicoidea (Polygyridae + (Camaenidae (Helicidae + Geomitridae))) [7, 86, 88]. Our results are consistent with both Helicoidea topologies depending on the phylogenetic approach used, although with some low support values. Although Xie et al. [86] suggest that the topology that places Polygyridae in a basal position would make more sense from a biogeographic perspective, the topological location of this group and the relationships among helicoids remain unclear. In this study, we observed different positions of the group formed by *N. nux* (Orthalicoidea) and *O. idahoensis* (Punctoidea) within "non-achatinoids", with low support values in both cases. The most recent phylogenetic studies with stylommatophoran mitogenomes were not concordant with respect to the relationships deduced between these groups. While the results of Xie et al. [86] grouped Orthalicoidea with Limacoidea, the results of Damatac and Fontanilla [88] grouped them with Succineoidea. The results of Linscott and Parent [89] grouped Orthalicoidea as follows: Punctoidea + (Orthalicoidea + (Succineoidea + Achatinelloidea)); however, the results of Doğan et al. [7] grouped Orthalicoidea + Arionoidea. Thus, this variation in the evolutionary affinities evidences an insufficient representativeness of the different taxonomic levels within Stylommatophora since the proposed relations are changing as new mitogenomes are made available. In addition, the BI analysis performed under the site-heterogeneous CAT-GTR model did not

improve the resolution of the trees inferred under the site-homogeneous models. Despite recovering all 11 superfamilies of Stylommatophora involved in this work as monophyletic groups, two nodes remained unresolved. Consequently, further research based on increased taxon sampling of Stylommatophora species is required to a better understanding of the evolutionary relationships within this group.

Among the additional phylogenetic markers provided by mitogenomes, several authors have mentioned the resolving potential of gene rearrangements [9, 90]. In this work we found the mitochondrial genome of *Omalonyx unguis* to present the same genetic arrangement as reported for *Succinea putris* [39]. To date and within Stylommatophora, Succineoidea is the only superfamily presenting the tRNA arrangements *tRNA-Pro–tRNA-Leu$_1$–tRNA-Ala* and *tRNA-Phe–tRNA-Asp–tRNA-Cys*. In addition, an inversion and transposition of *tRNA-Tyr–tRNA-Trp* was observed, and they were located between *tRNA-Ser$_2$* and *tRNA-Ser$_1$* (*tRNA-Ser$_2$–tRNA-Tyr–tRNA-Trp–tRNA-Ser$_1$*), with both genes encoded on the minus strand instead of the plus strand. While only two mitogenomes are available for the group, the reported changes seem to represent conserved arrangements that may constitute synapomorphies for Succineoidea. Further research based on more succinids are required to test and validate this assumption.

## Conclusions

This study provides the complete mitochondrial genome of the semi-slug *Omalonyx unguis* (Gastropoda: Succineidae), consisting of 13,984 bp and with a typical gene content. Genomic features were similar with those of other stylommatophoran mollusks. Additionally, we present, to our knowledge, the first structural model of the complete *12S-rRNA* (small-subunit rRNA) within Mollusca. The gene rearrangement was identical to that reported for *Succinea putris*. Both BI and ML analyses supported Stylommatophora monophyly with *Achatina fulica* at the basal bifurcation. The reconstructions within Stylommatophora were similar to those reconstructed in earlier studies. The arrangements *tRNA-Pro–tRNA-Leu$_1$–tRNA-Ala*, *tRNA-Phe–tRNA-Asp–tRNA-Cys* and *tRNA-Ser$_2$–tRNA-Tyr–tRNA-Trp–tRNA-Ser$_1$* (with *tRNA-Tyr* and *tRNA-Trp* on the minus strand) were present only in Succineoidea, and these arrangements are suggested to represent synapomorphies for the superfamily. Further research based in increased taxon sampling of succinids is required to confirm this hypothesis.

## Supporting information

**S1 Fig. The putative stem-loop structure that is found in the longest non-coding region, located between *cox3* and *tRNA-Ile* genes.**
(TIF)

**S2 Fig. Bayesian Inference tree under the site-heterogeneous CAT-GTR model based on the 13 PCGs dataset for stylommatophoran mollusks.** The trees were rooted with three outgroups (*Carychium tridentatum*, *Platevindex mortoni* and *Galba pervia*). The scale (0.6) shows evolutionary distances. Posterior probability support values are shown in the nodes. The GenBank accession numbers of the species are shown in Table 1; *Omalonyx unguis* was sequenced in this study.
(TIF)

**S1 Table. Individual-specific primers of *Omalonyx unguis*.** Primer sequences are in the 5' to 3' direction. The primers naming system follows that of White et al. [39]. The genes encoded in the minus strand are underlined.
(DOCX)

**S2 Table. Codon usage in *Omalonyx unguis*.** RSCU: relative synonymous codon usage. (DOCX)

## Author Contributions

**Conceptualization:** Leila Belén Guzmán, Ariel Aníbal Beltramino.

**Data curation:** Leila Belén Guzmán, Roberto Eugenio Vogler.

**Formal analysis:** Leila Belén Guzmán.

**Investigation:** Leila Belén Guzmán, Roberto Eugenio Vogler, Ariel Aníbal Beltramino.

**Methodology:** Leila Belén Guzmán.

**Project administration:** Leila Belén Guzmán, Ariel Aníbal Beltramino.

**Supervision:** Roberto Eugenio Vogler, Ariel Aníbal Beltramino.

**Validation:** Roberto Eugenio Vogler, Ariel Aníbal Beltramino.

**Visualization:** Leila Belén Guzmán.

**Writing – original draft:** Leila Belén Guzmán, Roberto Eugenio Vogler, Ariel Aníbal Beltramino.

**Writing – review & editing:** Leila Belén Guzmán, Roberto Eugenio Vogler, Ariel Aníbal Beltramino.

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
