## [Decision Letter · Decision Letter 0]

1 Apr 2021

PONE-D-21-05532

The mitochondrial genome of the semi-slug Omalonyx unguis (Gastropoda: Succineidae) and the phylogenetic relationships within Stylommatophora

PLOS ONE

Dear Dr. Guzmán,

Thank you for submitting your manuscript to PLOS ONE. After careful consideration, we feel that it has merit but does not fully meet PLOS ONE’s publication criteria as it currently stands. Therefore, we invite you to submit a revised version of the manuscript that addresses the points raised during the review process.

The reviewers were both impressed by the overall quality of the manuscript. That being said, they each provide useful and substantive feedback and suggestions that would improve the final product. Please pay particular attention to the phylogenetics suggestions, especially concerning model choice and LBA, and the issues with taxonomy. I agree with the reviewers that some of the figures are too difficult to interpret, so please try to improve the readability and descriptions thereof.

We look forward to receiving your revised manuscript.

Kind regards,

Michael Scott Brewer, Ph.D.

Academic Editor

PLOS ONE

Journal Requirements:

Additional Editor Comments (if provided):

Reviewers' comments:

Reviewer's Responses to Questions

**Comments to the Author**

1. Is the manuscript technically sound, and do the data support the conclusions?

Reviewer #1: Yes

Reviewer #2: Yes

2. Has the statistical analysis been performed appropriately and rigorously? 

Reviewer #1: Yes

Reviewer #2: Yes

3. Have the authors made all data underlying the findings in their manuscript fully available?

Reviewer #1: No

Reviewer #2: Yes

4. Is the manuscript presented in an intelligible fashion and written in standard English?

Reviewer #1: Yes

Reviewer #2: Yes

5. Review Comments to the Author

Reviewer #1: Comments for the author

The paper reports the sequencing and annotation of the mitogenome of a semi-slug Omalonyx unguis for the first time. The authors describe general features of the mitogenome, such as gene content and arrangements, nucleotide compositions and codon usage, secondary structures of the tRNA and rRNA genes as well as non-coding region. Additionally, the authors reconstructed the phylogeny of Stylommatophora using the concatenated amino acid sequences of mitochondrial PCGs of 38 species. The paper provides a valuable resource for understanding of the mitogenome structure of O. unguis and mitogenome evolution in the stylommatophoran species. The manuscript is also valuable in that the authors predict the secondary structure of 12S rRNA in Mollusca for the first time. The manuscript is clearly written, however I found some minor problems that would be better if solved:

• Line 141: The authors stated they performed Gblocks under relaxed settings and cited Castresana, 2000. However, in this article Castresana does not state which settings are accepted as relaxed. In my opinion, it would be better if they cite Talavera and Castresana, 2007 (doi: 10.1080/10635150701472164) or explain which parameters they used for “relaxed”.

• Line 237-239: The display of the rearrangements with the phylogenetic tree is nice and descriptive; however, I didn’t understand why they give this one sentence here. When reading, the pass between two subject had come abrupt to me and seem to be irrelevant. This part should be enriched.

• I could not understand why the author have preferred the ML phylogeny in Figure 7.

• I would prefer if the authors choose a sorting strategy in Table 1. It has categorized neither alphabetically nor by family. And, also it would be better, if they give the main species of the manuscript at the top of the table and categorize remainings up to families.

• In general, I find the figure legends poor. Legends must be understandable by themselves.

• As to phylogenies, unfortunately, I could not read the node labels in Fig. 6 because of the resolution. Therefore, I could not to pass judgment on related parts when authors stated as low support or high support.

• Finally, there was a systematic mistake about superfamilies. The species of Polygyra cereolus and Praticolella mexicana (Polygyridae) classified under Polygyroidea superfamily rather than Helicoidea in current sources (please see Bouchet et al., 2017; NCBI Taxonomy Browser). The phylogeny should be discussed again considering this classification.

Reviewer #2: The authors sequenced the mitochondrial genome of Omalonyx unguis. They annotated it and show the mitogenome features and was compared with remain mitogenomes of Stylommatophora. Also, the authors assembled a phylogenomic dataset based on 13 mitochondrial protein coding genes using this new mitogenome, giving the phylogenetic status of this genus. The MS was well written, the methods well executed and the results supported by the data. The conclusions were very well conducted.

Overall, the message present in this paper is straightforward and deserves to be published. The mitogenome of Omalonyx unguis increase the taxon sampling of the family, confirm its monophyly, give phylogenetic status of the genus, and confirm a molecular synapomorphy of Succineoidea (mitogenome gene rearrangement). However, there are few comments that need to be addressed.

Given the well-known acceleration of mutations rates expressed by heterobranch clades (and other land clades, e.g., Helicinoidea and Hydrocenoidea), which confer long branches in the trees of these clades. These long branches generate, potentially, biases in the phylogenetic reconstruction (by e.g., LBA, amino-acid composition). I suggest to the authors run the phylogenetic analyses using (at least one of two) heterogeneous models, as is the case of CAT model (implemented in phylobayes) OR the mixture model C10-C60 (implemented in IQ-TREE). These last models require less computational effort (using ultra-fast bootstrap) and run faster than CAT. These kind of substitution models have shown well performance avoiding phylogenetic biases in gastropod phylogeny (Uribe et al., 2019. Molecular phylogenetics and evolution, 133, 12-23) and are become in an inevitable analysis that allow know if the analyses are biased.

Finally, the phylogenetic trees could be represented in just one tree collapsing the nodes with low support. This may improve the figure and the two topologies could be disscused in the text and be showed as SM.

Please, the authors should describe the abbreviation used, e.g., transfer (t)RNA, ribosomal (r)RNA.

Figure 2. To makes this figure more informative the authors could display the biased codon in bold.

6. PLOS authors have the option to publish the peer review history of their article (what does this mean?). If published, this will include your full peer review and any attached files.

Reviewer #1: No

Reviewer #2: No

---

## [Author Response · Author response to Decision Letter 0]

6 May 2021

May 3, 2021

Dear Academic Editor

PLOS ONE

Ph.D. Brewer, Michael Scott

The MS was revised taking into account the comments suggested by you and the reviewers. The responses to the comments are detailed below.

Sincerely,

Lic. Leila B. Guzmán

----------

Academic Editor:

- Thank you for submitting your manuscript to PLOS ONE. After careful consideration, we feel that it has merit but does not fully meet PLOS ONE’s publication criteria as it currently stands. Therefore, we invite you to submit a revised version of the manuscript that addresses the points raised during the review process. The reviewers were both impressed by the overall quality of the manuscript. That being said, they each provide useful and substantive feedback and suggestions that would improve the final product. Please pay particular attention to the phylogenetics suggestions, especially concerning model choice and LBA, and the issues with taxonomy. I agree with the reviewers that some of the figures are too difficult to interpret, so please try to improve the readability and descriptions thereof.

*We thank the Editor for the comments on the manuscript. We have made changes in the revised manuscript for trying to address all the suggestions.

----------

Reviewer No. 1

*We thank Reviewer No. 1 for the positive comments and suggestions on the manuscript. We have worked on all the marked issues to improve the manuscript.

Comments:

- Line 141: The authors stated they performed Gblocks under relaxed settings and cited Castresana, 2000. However, in this article Castresana does not state which settings are accepted as relaxed. In my opinion, it would be better if they cite Talavera and Castresana, 2007 (doi: 10.1080/10635150701472164) or explain which parameters they used for “relaxed”.

*We agree. “Castresana, 2000” was replaced by “Talavera and Castresana, 2007”.

- Line 237-239: The display of the rearrangements with the phylogenetic tree is nice and descriptive; however, I didn’t understand why they give this one sentence here. When reading, the pass between two subject had come abrupt to me and seem to be irrelevant. This part should be enriched.

*We agree. We modified the manuscript for trying to address this point.

- I could not understand why the author have preferred the ML phylogeny in Figure 7.

*We choose to show the ML phylogeny in Fig 7 as the topology obtained from this analysis better agrees with the results published in previous works (e.g. Xie et al., 2019; Doğan et al., 2020) in comparison to the topology obtained from the BI analysis.

References:

Xie G, Köhler F, Huang XC, Wu RW, Zhou CH, Ouyang S, et al. A novel gene arrangement among the Stylommatophora by the complete mitochondrial genome of the terrestrial slug Meghimatium bilineatum (Gastropoda, Arionoidea). Mol Phylogenet Evol. 2019;135: 177-184. doi: 10.1016/j.ympev.2019.03.002.7.

Doğan Ö, Schrödl M, Chen Z. The complete mitogenome of Arion vulgaris Moquin-Tandon, 1855 (Gastropoda: Stylommatophora): mitochondrial genome architecture, evolution and phylogenetic considerations within Stylommatophora. PeerJ. 2020;8: e8603. doi: 10.7717/peerj.8603.

- I would prefer if the authors choose a sorting strategy in Table 1. It has categorized neither alphabetically nor by family. And, also it would be better, if they give the main species of the manuscript at the top of the table and categorize remainings up to families.

*We have modified Table 1 as suggested.

- In general, I find the figure legends poor. Legends must be understandable by themselves.

*We have improved the figure legends as requested.

- As to phylogenies, unfortunately, I could not read the node labels in Fig. 6 because of the resolution. Therefore, I could not to pass judgment on related parts when authors stated as low support or high support.

*The figures were uploaded in high resolution, however, the .pdf file generated by the journal´s system for revision decreased the images quality. We have tested to download the figs from the links to each figure in the .pdf file, and we could obtain the .tif files in high quality for better visualization.

- Finally, there was a systematic mistake about superfamilies. The species of Polygyra cereolus and Praticolella mexicana (Polygyridae) classified under Polygyroidea superfamily rather than Helicoidea in current sources (please see Bouchet et al., 2017; NCBI Taxonomy Browser). The phylogeny should be discussed again considering this classification.

*We have carefully revised the taxonomy following the Reviewer No. 1 suggestions. The taxonomy used in our work agrees with that of Bouchet et al. (2017) and MolluscaBase (https://www.molluscabase.org/), where "Polygyridae" belongs to the superfamily "Helicoidea" (https://www.molluscabase.org/aphia.php?p=taxdetails&id=993919; Bouchet et al., 2017: page 367). We have also verified the current treatment of the taxon “Polygyroidea”. Polygyroidea Pilsbry, 1924 corresponds to a valid mollusk genus within the family Megomphicidae (https://www.molluscabase.org/aphia.php?p=taxdetails&id=995468). We think that the confusion regarding the classification of Polygyra cereolus and Praticolella mexicana is derived from the taxonomical treatment provided in the "NCBI Taxonomy Browser", as the classification presented there is different from that of Bouchet et al. (2017) and MolluscaBase.

- Polygyra cereolus:

https://www.ncbi.nlm.nih.gov/Taxonomy/Browser/wwwtax.cgi?mode=Undef&id=339431&lvl=3&p=has_linkout&p=blast_url&p=genome_blast&keep=1&srchmode=1&unlock

- Praticolella mexicana:

https://www.ncbi.nlm.nih.gov/Taxonomy/Browser/wwwtax.cgi?mode=Info&id=882625&lvl=3&p=has_linkout&p=blast_url&p=genome_blast&lin=f&keep=1&srchmode=1&unlock

However, the NCBI taxonomy database clearly indicates on their site: "Disclaimer: The NCBI taxonomy database is not an authoritative source for nomenclature or classification". For the aforementioned reasons, we have decided to maintain the taxonomy as originally presented in our manuscript.

Reference:

Bouchet P, Rocroi JP, Hausdorf B, Kaim A, Kano Y, Nützel A, et al. Revised classification, nomenclator and typification of gastropod and monoplacophoran families. Malacologia. 2017;61: 1-526. doi: 10.4002/040.061.0201.

----------

Reviewer No. 2

*We thank Reviewer No. 2 for the positive comments and suggestions on the manuscript. We have worked on all the marked issues to improve the manuscript.

Comments:

- Given the well-known acceleration of mutations rates expressed by heterobranch clades (and other land clades, e.g., Helicinoidea and Hydrocenoidea), which confer long branches in the trees of these clades. These long branches generate, potentially, biases in the phylogenetic reconstruction (by e.g., LBA, amino-acid composition). I suggest to the authors run the phylogenetic analyses using (at least one of two) heterogeneous models, as is the case of CAT model (implemented in phylobayes) OR the mixture model C10-C60 (implemented in IQ-TREE). These last models require less computational effort (using ultra-fast bootstrap) and run faster than CAT. These kind of substitution models have shown well performance avoiding phylogenetic biases in gastropod phylogeny (Uribe et al., 2019. Molecular phylogenetics and evolution, 133, 12-23) and are become in an inevitable analysis that allow know if the analyses are biased.

*We have performed the suggested analysis in PhyloBayes with the CAT model, and the results were incorporated into the manuscript as supplementary material, as the new analysis does not provide a higher resolution than the original trees. We have introduced all the information regarding this new analysis using a heterogeneous model in the main document. 

- Finally, the phylogenetic trees could be represented in just one tree collapsing the nodes with low support. This may improve the figure and the two topologies could be disscused in the text and be showed as SM.

*We prefer to maintain the trees as originally presented for a better comparation of topologies obtained. Nonetheless, we have included the new tree derived from the BI analysis with the CAT model into supplementary material.

- Please, the authors should describe the abbreviation used, e.g., transfer (t)RNA, ribosomal (r)RNA.

*We agree, we have defined the abbreviations. 

- Figure 2. To makes this figure more informative the authors could display the biased codon in bold.

*We have performed the modifications suggested in Fig 2 to improve clarity.

---

## [Decision Letter · Decision Letter 1]

1 Jun 2021

PONE-D-21-05532R1

The mitochondrial genome of the semi-slug Omalonyx unguis (Gastropoda: Succineidae) and the phylogenetic relationships within Stylommatophora

PLOS ONE

Dear Dr. Guzmán,

Thank you for submitting your manuscript to PLOS ONE. After careful consideration, we feel that it has merit but does not fully meet PLOS ONE’s publication criteria as it currently stands. Therefore, we invite you to submit a revised version of the manuscript that addresses the points raised during the review process.

Please pay particular attention to the minor issues raised by the reviewer.

We look forward to receiving your revised manuscript.

Kind regards,

Michael Scott Brewer, Ph.D.

Academic Editor

PLOS ONE

Journal Requirements:

Reviewers' comments:

Reviewer's Responses to Questions

**Comments to the Author**

1. If the authors have adequately addressed your comments raised in a previous round of review and you feel that this manuscript is now acceptable for publication, you may indicate that here to bypass the “Comments to the Author” section, enter your conflict of interest statement in the “Confidential to Editor” section, and submit your "Accept" recommendation.

Reviewer #1: All comments have been addressed

Reviewer #2: All comments have been addressed

2. Is the manuscript technically sound, and do the data support the conclusions?

Reviewer #1: Yes

Reviewer #2: Yes

3. Has the statistical analysis been performed appropriately and rigorously? 

Reviewer #1: Yes

Reviewer #2: Yes

4. Have the authors made all data underlying the findings in their manuscript fully available?

Reviewer #1: Yes

Reviewer #2: Yes

5. Is the manuscript presented in an intelligible fashion and written in standard English?

Reviewer #1: Yes

Reviewer #2: Yes

6. Review Comments to the Author

Reviewer #1: I am happy to see the revised version of the manuscript and I think the authors made an excellent work with the revision of the manuscript. I’ve checked the revised version carefully. The authors answer and correct all the comments proposed by the reviewers in a clearly way. However, I just have a few minor suggestions, as I stated below which should be corrected.

-Lines 23-25: The sentence “Phylogenetic analyses based on the O. unguis mitogenome and 37 species of Stylommatophora were performed using amino acid sequences from the 13 protein-coding genes.” should be revised as “Phylogenetic analyses based on the mitogenomes of O. unguis and 37 other species of Stylommatophora were performed using amino acid sequences from the 13 protein-coding genes.” for the clarity.

-Line 32: “encodes 2 ribosomal RNAs” should be revised as “encodes two ribosomal RNAs”

-In the legend of Figure 7 at lines 273-274: I would prefer to see the first letter of three-letter coded amino acids in capital.

I hope my comments are clear and somehow helpful.

Reviewer #2: (No Response)

7. PLOS authors have the option to publish the peer review history of their article (what does this mean?). If published, this will include your full peer review and any attached files.

Reviewer #1: No

Reviewer #2: No

---

## [Author Response · Author response to Decision Letter 1]

1 Jun 2021

MS PONE-D-21-05532R1

Title: “The mitochondrial genome of the semi-slug Omalonyx unguis (Gastropoda: Succineidae) and the phylogenetic relationships within Stylommatophora”

June 1, 2021

Dear Academic Editor

PLOS ONE

Ph.D. Brewer, Michael Scott

The MS was revised taking into account the comments suggested by you and the reviewers. The responses to the comments are detailed below.

Sincerely,

Lic. Leila B. Guzmán

---

Academic Editor:

Thank you for submitting your manuscript to PLOS ONE. After careful consideration, we feel that it has merit but does not fully meet PLOS ONE’s publication criteria as it currently stands. Therefore, we invite you to submit a revised version of the manuscript that addresses the points raised during the review process.

Please pay particular attention to the minor issues raised by the reviewer.

-We have introduced the suggested changes in the revised manuscript.

Reviewer No. 1:

-We thank Reviewer No. 1 for the positive comments on the revised manuscript and the suggestions provided.

Comments:

Lines 23-25: The sentence “Phylogenetic analyses based on the O. unguis mitogenome and 37 species of Stylommatophora were performed using amino acid sequences from the 13 protein-coding genes.” should be revised as “Phylogenetic analyses based on the mitogenomes of O. unguis and 37 other species of Stylommatophora were performed using amino acid sequences from the 13 protein-coding genes.” for the clarity.

-We agree. We have incorporated the corrected sentence.

Line 32: “encodes 2 ribosomal RNAs” should be revised as “encodes two ribosomal RNAs”.

-We agree. “2” was replaced by “two”.

In the legend of Figure 7 at lines 273-274: I would prefer to see the first letter of three-letter coded amino acids in capital.

-We agree. We have added the first letter of three-letter coded amino acids in capital.

---

## [Editor Report · Decision Letter 2]

11 Jun 2021

The mitochondrial genome of the semi-slug Omalonyx unguis (Gastropoda: Succineidae) and the phylogenetic relationships within Stylommatophora

PONE-D-21-05532R2

Dear Dr. Guzmán,

We’re pleased to inform you that your manuscript has been judged scientifically suitable for publication and will be formally accepted for publication once it meets all outstanding technical requirements.

Kind regards,

Michael Scott Brewer, Ph.D.

Academic Editor

PLOS ONE
---

## [Editor Report · Acceptance letter]

18 Jun 2021

PONE-D-21-05532R2 

The mitochondrial genome of the semi-slug *Omalonyx unguis* (Gastropoda: Succineidae) and the phylogenetic relationships within Stylommatophora 

Dear Dr. Guzmán:

I'm pleased to inform you that your manuscript has been deemed suitable for publication in PLOS ONE. Congratulations! Your manuscript is now with our production department. 

Kind regards, 

on behalf of

Dr. Michael Scott Brewer 

Academic Editor

PLOS ONE